# Beyond Peat: Wood Fiber and Two Novel Organic Byproducts as Growing Media—A Systematic Review

**DOI:** 10.3390/plants14131945

**Published:** 2025-06-25

**Authors:** Anna Elisa Sdao, Nazim S. Gruda, Barbara De Lucia

**Affiliations:** 1Department of Soil, Plant and Food Sciences, University of Bari Aldo Moro, Via Amendola 165/A, 70126 Bari, Italy; 2Institute of Crop Science and Resource Conservation (INRES), Horticultural Sciences, University of Bonn, Auf dem Hügel 6, 53121 Bonn, Germany

**Keywords:** brewer’s spent grain, bioresources, circular economy, coffee silverskin, containerized cultivation peat alternative, renewable raw material, sawdust

## Abstract

Environmental concerns drive the search for sustainable organic alternatives in horticultural substrates. This review critically examines three agro-industry renewable byproducts—wood fiber, coffee silverskin, and brewer’s spent grain—as partial peat substitutes. We aimed to comprehensively analyze their origin, processing methods, current applications, and key physical, hydrological, and chemical properties relevant to horticultural use. In soilless culture, wood fiber can be used as a stand-alone substrate. When incorporated at 30–50% (*v*/*v*) in peat mixtures, it supports plant growth comparable to peat; however, higher proportions may restrict water and nutrient availability. Coffee silverskin demonstrates high water retention and nutrient content, but its inherent phytotoxicity requires pre-treatment (e.g., co-composting); at concentrations up to 20%, it shows promise for potted ornamental crops. Brewer’s spent grain is nutrient-rich but demands careful management due to its rapid decomposition and potential salinity issues; inclusion rates around 10% have shown beneficial effects. In conclusion, when used appropriately in blends, these bio-based byproducts represent viable alternatives to reduce peat dependence in vegetable and ornamental cultivation, contributing to more sustainable horticultural practices. Future research should optimize pre-treatment methods for coffee silverskin and brewer’s spent grain, investigate long-term stability in diverse cropping systems, and explore novel combinations with other organic waste streams to develop circular horticultural substrates.

## 1. Introduction

Soilless culture systems refer to all plant cultivation methods without using natural soil. These systems are generally classified into two main categories: liquid-based and solid media systems. Among liquid-based approaches, hydroponics is the most common, where plant roots grow directly in a nutrient solution that supplies water and essential minerals. In contrast, solid and aggregate media systems use substrates that provide physical support and a medium for delivering water and nutrients [1]. Here, we will use the terms “substrate” and “growing media” interchangeably.

Solid media systems typically arrange substrates in containers, foil-lined beds, rows, or shallow layers. One of the most significant applications of this method is in the production of ornamental plants. The containerized cultivation of trees, shrubs, and herbaceous species has become a central practice in the decorative sector, offering flexibility, efficient space utilization, and better environmental control. Containerized ornamental plants are used in urban and rural environments, improving aesthetic appeal, supporting biodiversity, and contributing to human well-being [2].

A growing medium placed in a container provides anchorage for roots while maintaining suitable levels of water and air to support healthy plant growth. However, container systems inherently limit root expansion, gas exchange, and water buffering capacity. To overcome these constraints, substrates typically comprise blended organic and inorganic components that balance the physical and chemical characteristics needed for optimal plant performance [3,4,5].

Inorganic materials commonly used in substrate mixtures include both natural and processed materials. Natural materials, such as sand, pumice, and tuff, provide good drainage and aeration. On the other hand, processed materials like perlite, vermiculite, rock wool, expanded clay, and zeolite are valued for their structural consistency and long-term stability [4,6,7]. Although they often comprise a smaller portion of substrate blends [8], these materials remain essential for ensuring reliable performance, especially in intensive horticultural systems [9].

Increasingly, there is a shift toward sourcing substrate components locally and sustainably. Renewable and regionally available materials are encouraged to reduce environmental impact and production costs while promoting circular practices in horticulture [7,8,10]. Consequently, the careful selection and formulation of substrates are crucial for the success of soilless cultivation systems, particularly in high-value sectors such as ornamental plant production.

Peat is the most widely used material in horticultural substrates [8,11]. From an agronomic perspective, it is considered an ideal substrate for propagation and cultivation due to its ability to retain water and nutrients and its porosity, which provides oxygen to the roots and facilitates drainage. Adding specific amendments can easily correct its low pH and nutrient content [11].

However, the use of peat in horticulture has been strongly criticized because of environmental and climate change concerns [7] and because it is a non-renewable organic matrix [12]. The areas where peat accumulates over time, called peatlands, cover an estimated 4 million km^2^, which accounts for 2.8% of the Earth’s surface. It is a unique ecosystem that develops in oxygen-deficient environments with groundwater or heavy precipitation, forming carbon- and nitrogen-rich organic soils. Their carbon storage capacity is significant, estimated at approximately 644 Gt, accounting for 21% of global organic carbon [13]. Therefore, peatlands are crucial to climate stability [14]. Also, the “2030 Agenda for Sustainable Development" in target 15.1 refers to “Protect, restore and promote the sustainable use of the earth’s ecosystem”, among which the protection of peatlands as environments rich in biodiversity and useful for CO_2_ sequestration [15].

In *Pinus engelmannii* Carr. (Apache Pine) seedling production [16], the substrate is a critical factor, as the properties of its constituent materials significantly influence growth and future plant development. In general, the root system of the seedlings produced in sawdust-based growing media presented good physiognomy based on the points of growth observed in the root, possibly due to the ratio of the dry root biomass and the growth potential.

Moreover, the organic and inorganic components also directly impact production costs. González-Orozco et al. [16] pointed out the difference in cost of USD 2.34 between 50% composted pine bark (USD 3.49) and 70% pine sawdust (USD 1.15) peat-based growing media. Therefore, the substrate’s cost has led to research on replacing peat with local materials.

The physical properties of growing media depend on the morphological characteristics of the particles composing them [17]. A growing medium must strike a balance between water retention, air supply, and drainage to prevent waterlogging and disease, while ensuring sufficient porosity for root aeration. In addition, it must guarantee a stable structure over time, resistant to shrinkage or expansion [18]. Chemically, the substrate should have a neutral to slightly acidic pH [19], low salinity, and a slow decomposition rate, and contain essential nutrients or allow for adequate fertilization [18]. Other desirable qualities include the absence of weed seeds, nematodes, pathogens, and phytotoxins [20]. From a practical perspective, the material used for the substrate should be readily available, low-cost, and easy to mix with different matrices [5,7,8].

Agriculture-based industries generate approximately 1.3 billion tons of waste annually [21,22]. Most of this waste is disposed of through incineration or landfilling, which presents considerable environmental, economic, and social challenges. Capanoglu et al. [23] emphasize that the carbon footprint associated with agro-industrial waste is significant, contributing approximately 3.3 billion tons of CO_2_ emissions to the atmosphere each year. This excessive generation of agro-industrial waste disproportionately impacts developing countries, where viable alternatives for recovering and reusing these valuable resources remain limited [24], underscoring a critical gap in sustainable waste management.

Forestry byproducts are residual logging and wood processing materials, including bark, branches, sawdust, and industrial wood residues. Agro-industry byproducts are materials left over after processing agricultural goods and can be valuable rather than wasteful. Often considered waste, they hold potential for bioenergy, bioproducts, and sustainable materials, supporting circular economy approaches while reducing the environmental impact of forest-based industries [25]. Utilizing food, drink, and agricultural waste presents both a challenge and a necessity in daily life. This aligns with the core concepts of the circular economy, which are based on the three rules: Reduce, Reuse, and Recycle. Furthermore, target 12.5 is a key component of Sustainable Development Goal 12, which focuses on achieving sustainable consumption and production patterns. It emphasizes moving from a “take, make, and dispose” linear economy towards a more circular one [4,26].

## 2. Organic Byproducts as Growing Media

Organic byproduct sources derived from forestry and agro-industrial processes—such as wood fiber, coffee silverskin, and brewer’s spent grain—are gaining attention for their potential use in horticultural growing media (Figure 1).

Several standard growing media contain up to 30 to 40% by volume of wood fiber (WF), and its potential for co-use in growing media has not yet been fully exploited. Coffee silverskin (CS), a thin tegument directly covering the coffee bean, is a byproduct of the coffee roasting process. Considering the large quantities produced globally and their potential environmental impact if not effectively utilized, developing sustainable applications for CS becomes an apparent necessity. Similarly, brewer’s spent grain (BSG), the solid residue remaining after brewing, stands out as an attractive and cost-effective raw material. Its year-round availability and rich composition of carbon and nitrogen sources make it particularly appealing for diverse agricultural and horticultural applications.

Despite the recognized potential of these byproducts, a comprehensive synthesis of their diverse properties and efficacy in horticultural applications remains a critical research gap. While wood fiber has established its presence as a growing media component, its full potential for co-utilization with other novel organic byproducts, especially in diverse horticultural contexts, is still underexplored.

This systematic review addresses this gap by examining three organic byproducts, two of which are unconventional—coffee silverskin and brewer’s spent grain—alongside wood fiber, all sourced from the forestry and agro-industrial sectors, for their potential use in growing media. The objective is to provide a comprehensive overview of their origin, processing, current applications, and key chemical and hydrological properties, focusing on their individual and synergistic effectiveness in supporting the cultivation of a wide range of horticultural crops.

## 3. Methodology

### 3.1. Formulation of the Problem

The use of peat in horticultural substrates raises environmental and climate change-related concerns. Therefore, it is urgent to identify organic replacement byproducts that can be used in substrate composition and validated in cultivation.

### 3.2. Keyword Identification

Thomson Reuters’ Web of Science, Elsevier’s Scopus, and Google Scholar were queried from 1972 to April 2025 for worldwide published scientific publications, written in English, with subject areas limited to Agricultural and biological sciences, and Environmental science and they were identified using the following keywords: (i) potted growing media (Figure 2a); (ii) agro-environmental sustainability (Figure 2b); (iii) forestry and agro-industry byproducts (Figure 2c); (iv) unconventional organic matrices (Figure 2d). Subsequently, the process of identification of bibliographical references took place by title/keywords/abstract and discarding those irrelevant.

About 67% of the references are categorized as organic and inorganic, peat-free, and sustainable growing media (Figure 2a). Regarding agro-environmental sustainability (Figure 2b), 30% of the references are placed in themes related to peatland C-rich ecosystems, and 30% in sustainable development. In Figure 2c (forestry and agro-industry byproducts), the agro-industry byproducts account for 66% of the references. Figure 2d (unconventional organic matrices) shows that wood fiber (WF) references consist of 28%, those of coffee silverskin (CS) of 38%, and those of brewer’s spent grain (BSG) of 34%.

Figure 3 shows the references (%) of agricultural use: in WF (a), 91% represents ornamentals and vegetables; in CS (b), soil amendments and compost represent 49% of uses, while peat replacement reaches 17%. The BSG (c) agricultural use consists of 44% as fertilizer.

## 4. Wood Fiber

Wood-based growing media, including bark, chips, biochar, sawdust, and wood fibers, have been used in Europe for a long time. Their market is expected to reach 30 Mm^3^ year^−1^ in 2050 [27]. Wood fiber (WF) is generally made from lumber, and as a substrate, it contains little or no bark [28,29]. It is considered a renewable resource that can be obtained from sustainably managed forests, thus, as an organic material resulting from industrial wood processing. Production typically involves subjecting wood to high pressure and temperature. Wood fiber is often processed through grinding, where the size of the wood pieces is reduced and the fibers separate from each other. It appears as filaments. It is predominantly sourced from coniferous species such as Norway spruce (*Picea abies* (L.) H. Karst.), pine (*Pinus* spp.), lodgepole pine (*Pinus contorta* Douglas ex Loudon), or broadleaf trees such as chestnut (*Castanea sativa* Mill.) [30]. In the late 1980s, the first to grind pine wood for use as a soilless culture substrate were Laiche Jr. and Nash [31]. Following their promising results, interest in commercial WF grew in the early 1990s. In 1994, Benoit and Ceustermans [32] described WF in cultivation as the phenomenon of degradation by microorganisms, which leads to a worsening of the physical properties and nitrogen immobilization [33,34,35,36]. In Europe, WF growing media were particularly popular in Germany in the early 2000s, as reported by Gumy [37] and Bobo and Jackson [38]. In the same period, they also attracted the attention of the USA [8].

### 4.1. The Production Process

Wood fiber (WF)-based growing media production involves turning raw wood, often a byproduct of sawmilling or wood processing, into a stable, safe, and horticulturally suitable substrate. The process varies depending on the desired fiber quality and end use. Wood is mechanically defibered using disc refiners, hammer mills, or high-speed grinders [39], and then treated with steam at temperatures between 80 °C and 120 °C (stabilized WF), ensuring the elimination of plant pathogens, reducing the resin content (which could be phytotoxic), and partially stabilizing the material biologically.

WF can be produced in both coarse and fine grades, and the characteristics of the source material and fiber structure are significant in plant cultivation. During this process, nitrogen (N) is often used to enrich it, particularly when derived from wood chips or bark, and tends to have a relatively low N content [28]. The most commercially available WF products have included various trade names over the years, although some are no longer actively produced or widely distributed. Notable examples include Toresa, Hortifibre, Cultifibre, Piatel, Torbella, SunFiber Nuova Flesan, Ecofibrex, and Brill Substrate Fiber.

### 4.2. Agricultural and Non-Agricultural Uses

In farming systems, wood fiber is used to cultivate vegetables [40,41,42,43], ornamentals [12,44,45,46], forest seedlings [47], and trees [48]. WF use is consolidated above all in the production of insulating panels for construction and laminates [25], as a mechanical buffer, and electrolyte reservoir for sodium–ion batteries [49].

### 4.3. Physical, Hydrological, and Chemical Characterizations

#### 4.3.1. Physical and Hydrological Characterizations

Gruda and Schnitzler [50,51] found a low bulk density, good drainability, and rewettability of wood fiber (WF). Domeño et al. [52] describe a bulk density of 0.07 g cm^−3^ (recommended range (R.R.) < 0.4) and a particle density of 1.56 g cm^−3^ (R.R. = 1.4–2.6). Commercial products are typically mixtures of particle size fractions in varying percentages.

Wood fiber (WF) moisture content is between 10.3% [53] and 41% [41]. Gruda and Schnitzler [50] reported that pure or 50–70% wood fiber: white peat (*v*:*v*) showed a similar total pore space to each other and in line with the R.R. >85% found by De Boodt and Verdonck [54]. These WF-based growing media showed lower readily available water (E.A.W. calculated as the amount of free water of the tested materials when the water tension increased from pF = 1.0 to 1.7) than peat, according to Domeño et al. [52], who report an E.A.W. of 13.8 vol. %. In mixtures of peat and WF (*v*:*v*, 50:50), the readily available water reached 25.1%, which is lower than that of peat (33.5%) but is still within the ideal range of 20–30%, as noted by De Boodt and Verdonck [54] (1972), with rapid drainage of surplus water.

Reineke et al. [45] reported that in petunia, the thermo-hydrolytically and thermally treated WF, at 30–60% *v*/*v*, decreased shoot biomass and flower number with increasing WF content due to lower water-holding capacity. Therefore, the WF proportion and irrigation regime are key to achieving a peat substitution without compromising quality.

Finally, wettability characterization, defined by Michel [55] as the degree of hydrophilicity, is particularly crucial in containerized crops, in which fluctuations can quickly lead to oxygen stress and root asphyxiation in cases of excessive irrigation, or conversely, to water stress and potential hydrophobicity when water inputs are insufficient. Barrett et al. [4] argued that WF exhibits a high degree of wettability: in fact, it is used, in blended growing media, to optimize physical properties, reduce bulk density, increasing air space and, thus, improving re-wetting capacity; for the same reason, it is not used as a stand-alone component for the constitution of a substrate [56].

Table 1 reports the main physical and hydrological properties determined for wood fiber, as described in previous studies.

#### 4.3.2. Chemical Characterization

Regarding the chemical properties, wood fiber has a subacid pH, between 4.1 [53] and 6.0 [30], and a variable but still very low EC: from 4.0–6.0 [30] to 18.6 [57] to 28 mS cm^−1^ [53]. This wide variability is due to the heterogeneity of the raw wood material. The ash content was estimated at 0.15% (*w*/*w*) [53]. Domeño et al. [52] stated a cation exchange capacity of 0.22 meq g^−1^; Zaccheo et al. [30] reported a total organic matter content of 95–98%, based on the findings of Zawadzińska et al. [53], along with an organic carbon content of 64–67%. Zaccheo et al. [30] estimated the total nitrogen content to be 0.20–0.40%, consistent with Zawadzińska et al. [53], who found 0.16%. Čepulienė et al. [41] identified 0.22%, while Sdao et al. [57] measured 0.45%. The C/N ratio noted by Zaccheo et al. [30] of 168–336 differs little from that observed by Čepulienė et al. [41] of 435 but is in line with the value documented by Domeño et al. [52], equal to 456. Wood fiber is primarily made of cellulose and lignin, rich in C and poor in nitrogen. Čepulienė et al. [41] recorded a P content of 57 mg kg^−1^ and a K of 472 mg kg^−1^. Finally, Zawadzińska et al. [53] indicate Ca = 1574 mg kg^−1^, Mg = 346 mg kg^−1^, and Fe = 46.5 mg kg^−1^. Maher et al. [28] determined a lignin content of 360 g kg^−1^ (dry weight), while Domeño et al. [52] quantified a cellulose content of 180 g kg^−1^ (dry weight) and a hemicellulose content of 489 g kg^−1^ (dry weight). Table 2 summarizes the key chemical characteristics of wood fiber, including C/N ratio and pH, based on the relevant literature.

#### 4.3.3. Practical Applications

Wood fiber’s low bulk density, good trainability, and high wettability make it an excellent component for lightening growing media and enhancing aeration in containerized crops, particularly when blended with peat. While its lower readily available water capacity and high C/N ratio necessitate careful irrigation management and potential nitrogen supplementation, WF-based mixtures optimize physical properties and improve re-wetting capacity, positioning wood fiber as a valuable partial peat substitute for diverse horticultural applications.

### 4.4. Crop’s Performance

In recent years, wood fiber (WF) has gained popularity and become a renewable raw material in peat-based substrates [58,59,60]. Studies on vegetables and ornamentals have shown differences in performance and fertilizer requirements, so the substrate composition needs to be optimized.

#### 4.4.1. Wood Fiber Peat Replacement up to 50%

In *Leucanthemum vulgare* [12], wood fiber (WF), at moderate levels (30–40% *v*/*v*) and combined with coir and compost, supported plant growth and nutrient uptake comparable to, or better than, peat-based control. Beretta and Ripamonti [61] report that a peat substitution rate of between 30% and 50% showed promising results in the cultivation of *Juniperus communis* ‘Repanda’, *J. communis* ‘Prince of Wales’, and *Ribes uva-crispa* ‘Hinnonmaki Rot’. In *Petunia* × *hybrida*, peat-reduced mixes up 35% WF yielded comparable plant vitality and flowering to peat control. In basil, the buffering capacity of WF may help maintain more stable pH levels [44]. Laun et al. [62] also stated that wood fiber in vegetable seedlings could replace up to 30 to 50% of peat volume.

#### 4.4.2. Wood Fiber Peat-Replacement Beyond 50%

Substrates containing 70% wood fiber (WF) led to significant biomass reduction and impaired performance, highlighting that excessive doses may compromise water retention and nitrogen availability [61]. Woznicki et al. [63] (2024) showed that substrates with high WF content were less acidic and rapidly increased pH in petunia cultivation, resulting in iron malabsorption. In Petunia and basil, WF ≥ 75% led to chlorosis, mainly growth reduction, unless pH and nutrition supply were taken care of. In the hydroponic strawberry system [42], the 100% spruce fiber (*Picea abies*) significantly advanced the ripening and commercial marketable yield, compared to the control peat and perlite. Similar results were achieved in studies by Gruda and Schnitzler [29] with lettuce transplants. Many authors agree that WF in the hydroponic cultivation of strawberries has given good results as a stand-alone substrate [42,64,65,66].

#### 4.4.3. Wood Fiber and Fertilization

Incorporating wood fiber components into peat-based soilless substrates can increase the amount of fertilizer N needed during production as a result of greater N immobilization [5,67]. Čepulienė et al. [41] conducted a study to explore the effect of WF as an alternative to peat in growing media for cucumber (*Cucumis sativus* L.) cultivation, focusing on its impact on nutrient content and uptake. This experiment compared different growing media combinations of peat and WF (0, 25, 50, 100 WF) with additional nitrogen fertilizer (13, 23, 30 g plant^−1^). Results showed that in the substrates with 50 and 100% WF content and with the increased additional fertilization, there was a significant reduction in dry biomass of the aerial part compared to the peat-based substrate without fertilization.

## 5. Coffee Silverskin

Coffee is among the most widely consumed beverages globally [68]. Coffee has become a beloved staple in people’s lives worldwide; over 2.25 billion cups of coffee are consumed worldwide each day [69]. Cultivation mainly occurs within the “coffee belt” between the Tropics of Cancer and Capricorn, at 25° North and 25° South latitudes. In the coffee industry, a “bag” refers to a standard unit of measurement for coffee, typically 60 kg of raw coffee beans, from Central and South America, accounting for 96.93 million bags, followed by Asia, with 41.91 million bags. Additional contributions came from Africa: 15.0 million bags. The global green coffee market was valued at approximately USD 39.97 billion in 2024 [70].

There are two main coffee species: *Coffea arabica* cultivated in Latin America and Ethiopia, accounting for 57,4% of production (102.2 million bags), and *C*. *canephora* (syn. *C.*
*robusta*) cultivated in Southeast Asia and Africa, which constitutes 42,6% (75.8 million bags) [71]. *C*. *arabica* grows at higher altitudes (usually 600–2000 m) and requires cooler temperatures. Arabica plants are more delicate and require more agronomic maintenance; *C*. *robusta* grows at lower altitudes, is more resistant to heat and drought, and is easier to cultivate [72]. Green coffee, along with its byproducts and residues, represents a valuable source of macromolecules (e.g., carbohydrates, proteins, pectin) and bioactive compounds. These can be cost-effectively extracted and repurposed for novel applications in the food, nutraceutical, agricultural, cosmetic, and pharmaceutical sectors, aligning with FAO’s bioeconomy and circular economy principles [73].

### 5.1. The Production Process

The *testa* surrounds the coffee bean, a thin, fibrous, and protective layer, often referred to as the silverskin, due to its silvery appearance. Once the green beans are picked and processed, the silverskin is typically eliminated during roasting, leaving the roasted beans used to make coffee [74]. Coffee silverskin (CS) represents the only byproduct released during the coffee roasting process, accounting for approximately 4.2% (*w*/*w*) of seed total weight. About 400,000 tons of this waste is generated each year [75,76]. For coffee producers, managing this waste represents a significant cost [77], and its storage also poses a risk, as CS is highly flammable [78].

### 5.2. Agricultural and Non-Agricultural Uses

Although coffee silverskin (CS) is primarily considered industrial waste, it is a biomass with a valuable raw input into a wide range of potential applications [79].

The favorable composition of coffee processing waste, its high availability, and low cost make it an excellent source of various bioactive compounds with nutritional value [80].

In recent times, the Farm-to-Fork strategy [81], Salbitani et al. [82], and Carnier et al. [83] have shown the use of CS for organic fertilization purposes due to its high content of macro and micronutrients. Other uses in agriculture have also been evaluated: Picca et al. [84] and González-Moreno et al. [85] evaluated their use in compost and vermicompost. Finally, Sdao et al. [57] evaluated its use as a substitute matrix for peat in containerized ornamental plants.

Regarding the non-agricultural uses, CS has been studied in the food sector [86] to enrich preparations such as cookies, chocolate cake [87], and candies, as a fat replacer for chicken hamburgers, and as a source of antioxidants [88]. Industrially, it can be used as a material for polymers [89] and packaging [90].

In cosmetics utilization, dos Santos et al. (2021) [91] showed that CS in sunscreen is promising, as is in other cosmetic applications [92]. CS has also been studied as a feed for another critical sector, such as insect breeding [93].

### 5.3. Physical, Hydrological, and Chemical Characterization

#### 5.3.1. Physical and Hydrological Characterization

Prandi et al. [94] describe the structural characterization of coffee silverskin (CS), supplied by Illycaffè S.p.A. (Trieste, Italy), as consisting of flakes ranging in size from 2 to 5 mm, with a bulk density of 0.18 g cm^−3^ and a particle density of 0.71 g cm^−3^. Moisture content ranged between 7.3 and 7.1, in agreement with Pourfarzad et al. [95] and Arya et al. [96]. Ballesteros et al. [97] report a water holding capacity of 5.11 g H_2_O g^−1^ dry sample, probably due to the high fiber content [96]. Sdao et al. [57] report a water retention of 91.4%. Table 3 presents the physical and hydrological parameters of coffee silverskin.

#### 5.3.2. Chemical Characterization

In coffee silverskin (CS), the pH, determined by Sdao et al. [98], was found to be 5.7; this value is consistent with that reported by Thligene et al. [99] of 5.6 and Del Pozo et al. [78] of 5.3. Electrical conductivity (EC), as reported by Sdao et al. [57], was measured at 1.8 dS m^−1^, which is in agreement with González-Moreno et al. [85]. However, this contrasts with the findings of Carnier et al. [83], who reported a significantly higher EC of 4.9 dS m^−1^. This discrepancy underscores the necessity of monitoring soil conditions during the application of CS to prevent salinization. The ash content, evaluated by Sdao et al. [98] with 6.6, aligns with the range reported by Pourfarzad et al. [95], Ballesteros et al. [97], and Arya et al. [96] of 5–7 g 100 g^−1^ dry weight.

The total organic matter content, as reported by Sdao et al. [98], was found to be 93.4 g 100 g^−1^, consistent with the 91.3 g 100 g^−1^ reported by González-Moreno et al. [85]. Carnier et al. [83] reported 43.3 g 100 g^−1^. The C/N ratio also varies: González-Moreno et al. [85] reported a minimum value of 11, while Ballesteros et al. [97] reported 14.4, and Carnier et al. [83] reported 19.1. The total Kjeldahl nitrogen is reported by Carnier et al. [82] as 0.14 g 100 g^−1^; in contrast, González-Moreno et al. [85] reported 5.1 g 100 g^−1^, and Sdao et al. [57] 2.39 g 100 g^−1^.

Sdao et al. [57] reported a total phosphorus content of 0.76 g 100 g^−1^. The presence of a considerable concentration of potassium, equal to 7,6 g kg^−1^, in this waste makes it particularly interesting for tropical soils, which are often poor in K, and many crops. Carbohydrates, including cellulose, hemicellulose, and lignin, were also analyzed. Arya et al. [96] assessed the hemicellulose and cellulose contents of 62.1 and 23.7 g 100 g^−1^; they also reported a lignin content of 28.6 g 100 g^−1^, which agrees with the findings of Mussatto et al. [100], who documented 30.2 g 100 g^−1^. Finally, protein content was determined by Arya et al. [96] to be 18.7 g 100 g^−1^, aligning with 16.2 g 100 g^−1^ reported by Mussatto et al. [100].

Coffee silverskin is rich in inhibitory substances, such as caffeine, tannins, and polyphenols, which are responsible for insufficient nitrogen mineralization. Consequently, while these residues are rich in organic matter and exhibit promising soil conditioning properties, their use as primary nitrogen sources is not recommended. Table 4 outlines the main chemical properties of coffee silverskin, including organic matter content and nutrient profile.

#### 5.3.3. Phytotoxicity

Another fundamental aspect of coffee silverskin (CS) is phytotoxicity. Picca et al. [84] showed that it is due to high polyphenols and chlorogenic acid content, which can limit its direct application as a soil amendment, despite its beneficial agronomic properties, such as high nitrogen and potassium content, and excellent water retention capacity. The co-composting CS with carbon-rich materials eliminates phytotoxicity while maintaining its agronomic benefits. The antioxidant properties of CS derive from its polyphenolic content: Ziemah et al. [101] report a total polyphenol content (GAE) of 140 mg g^−1^. Al-Charchafchi et al. [102] results indicate that chlorogenic acid possesses phytotoxic properties, as it negatively affects seed germination.

#### 5.3.4. Practical Applications

Owing to its high water holding capacity and significant organic matter content, coffee silverskin is a promising amendment for improving soil structure and water retention in growing media. However, its inherent phytotoxicity and potential for variable electrical conductivity (EC) and nitrogen immobilization primarily necessitate its use after co-composting carbon-rich materials. When properly treated, its potassium richness makes it particularly beneficial for K-deficient soils or specific crop requirements.

### 5.4. Crop’s Performance

Coffee silverskin (CS) was used as organic fertilizer, compost, vermicompost, and peat-based growing medium to cultivate herbaceous and ornamental crops.

#### 5.4.1. Coffee Silverskin as Organic Fertilizer

Salbitani et al. [82] show that at a 2% dose, CS improves soil properties and promotes barley growth and physiology. In comparison, at a 10% dose, it negatively impacts leaf proteins. It reduces Chl b and Chl a/Chl b ratios, probably due to potential disruption of the photosynthetic apparatus, decreasing chlorophyll synthase activity, or nitrogen availability. Carnier et al. [83] investigated CS. They experimented with coffee powder for maize (*Zea mays* L.) at the dose of 450 mg of Kjeldahl nitrogen per pot, compared to a control fertilized with ammonium nitrate. Maize growth in soil treated with coffee residues did not significantly differ from the control but was lower than in ammonium nitrate-treated plants, suggesting limited nitrogen availability.

#### 5.4.2. Coffee Silverskin as Co-Composting Matrix

Composting organic residues for agricultural soil amendment offers significant benefits. Through enhanced water retention, composting reduces inorganic fertilizer demand and irrigation requirements. This positions composting as a cost-effective and industrially scalable technology for organic waste recycling, yielding a valuable soil conditioner with fertilizing properties [103].

Picca et al. [84] demonstrated that coffee silverskin (CS) co-composting with gardening pruning waste and biochar produced nutrient-rich, non-phytotoxic compost with phytostimulant properties. González-Moreno et al. [85] examined the vermicomposting process of CS and spent coffee grounds (SCG), mixed with horse manure (HM). The vermicomposting was conducted at varying ratios of CS/SCG/HM (25, 50, 75, and 100%) and lasted 60 days. The biological results showed that the growth and reproduction of the worms were significantly affected by the different waste combinations. The total biomass growth was highest in CS = 50, while treatment with 100% CS showed the lowest growth. The worm growth rate followed a similar trend. Regarding *Triticum aestivum* seed germination, used as a compost maturity and non-phytotoxicity indicator, no significant differences were found between the various mixtures. However, the germination, at a dose of CS 50%, showed the best results.

#### 5.4.3. Coffee Silverskin as Peat-Based Growing Medium

Sdao et al. [57] evaluated the coffee silverskin (CS) at 0, 10, 20, and 40% doses in seedlings and sale pot plants, as a partial peat substitute in ornamental marigold (Tagetes patula) and nasturtium (Tropaeolum *minus*) cultivation. In marigold seedlings, the peat-based substrate achieved the highest chlorophyll content; the values decreased as the CS dose increased. In nasturtium, however, the chlorophyll content at the CS 10% dose was comparable to that of the control, exhibiting the highest values. As sale plants, the dry weight in tagetes did not show significant differences between treatments; in nasturtium, 20% CS reached results comparable to peat.

## 6. Brewer’s Spent Grain

Beer is one of the oldest and most popular alcoholic beverages in the world. According to the 2024 European Beer Trends Statistics Report [104], beer production reached 346 million hL in 2023, marking a decline of more than 3% compared to 2022. Italy ranked ninth in beer production, with a total output of 17,430 thousand hL.

### 6.1. The Production Process

Brewer’s spent grain (BSG) is the primary byproduct of the brewing industry, accounting for 85% of total brewing residues [105], with a global annual production of 12 million t. In producing the most common beer type, light lager, malt consumption is estimated at approximately 20 kg hL^−1^ [106,107]. BSG is made through a fermentation process in which the sugars contained in malt are transformed into alcohol and carbon dioxide through the action of yeasts. During the malting phase, barley grains are germinated and dried to develop the necessary enzymes for starch conversion. In mashing, the malt is mixed with hot water, allowing the enzymes to break down starches into fermentable sugars; this stage generates BSG [108]. The brewing industry significantly impacts the environment by generating substantial waste. Per 1000 tons of beer, approximately 137–173 tons of solid waste are produced, with spent grain accounting for roughly 20 kg per 1 hL of beer [109].

### 6.2. Agricultural and Non-Agricultural Uses

Traditionally considered waste, brewer’s spent grain (BSG) is now the focus of a few research studies aimed at developing sustainable strategies for its valorization. Several studies highlight its potential applications across various sectors [110]. BSG has also been studied and applied in agricultural uses, such as producing biochar [111], organic soil amendment [112], compost [113], and peat replacement [57]. 

Regarding non-agricultural uses, brewers’ spent grain (BSG) for animal feed is already consolidated and favored by breweries because it is environmentally sustainable and economically favorable [114]. BSG is used in the food industry for bakery products [115]; in plant-based meat [116], pasta [117], and yogurt [118], due to its high content of fiber, protein, and bioactive compounds. Moreover, it is utilized in the production of bioplastics and bio-composites [119], bioenergy [120], and reducing sugars for biofuels [121]. In recent years, the use of BSG in the medical-pharmaceutical field has been interesting and increasingly studied by researchers [122]. In animal nutrition, BSG supplementation has improved growth performance, immune responses, and antioxidant capacity, without detrimental effects on metabolic profiles [123,124].

### 6.3. Physical, Hydrological, and Chemical Characterization

Brewer’s spent grain (BSG) is a complex lignocellulosic material composed of the seed coat, pericarp, and husk layers of barley grain [125].

#### 6.3.1. Physical and Hydrological Characterization

Naibaho et al. [126] reported a bulk density of 0.129–0.159 g cm^−3^. Meneses et al. [127], Castro et al. [128], Cacace et al. [112], and Belardi et al. [129] agree on a moisture ranging between 78% and 84%. This high moisture content and fermentable sugars, mainly maltose and maltotriose [130], make BSG highly susceptible to microbial growth, leading to rapid spoilage within a short storage period of 7 to 10 days. Refrigeration or drying is advised to maintain physico-chemical properties and prevent deterioration [131]. Sdao et al. [57] report a water retention of 207%. Table 5 displays brewers’ spent grain’s principal physical and hydrological features, focusing on structure, moisture retention, and air capacity.

#### 6.3.2. Chemical Characterization

Brewer’s spent grain (BSG) pH is sub-acidic, with attested values of 5.5 [128], 4.5 [112], and 6.0 [98]. Sdao et al. [57] evaluated the electrical conductivity (EC) as 3480 μS cm^−1^. The ash content is variable, and it was accounted for in 2021 by Castro et al. (2.3%) [128], in 2023 by Belardi et al. (3.5%) [129], in 2024 by Mainali et al. (8.5%) [132], and in 2025 by Garcia et al. (1.9%) [133]. Sdao et al. [57] evaluated the Organic Matter content as 93.8 g 100 g^−1^. Organic Carbon content ranged between 47.3 [132] and 53% [134], and C/N ratio ranged from 10.8 [132] to 13.9 [134]. Nitrogen content ranged between 3.8 [134] and 5.1% [132].

According to Meneses et al. [127], the most abundant elements in BSG are phosphorus (P = 6000 mg kg^−1^), calcium (Ca = 3600 mg kg^−1^), magnesium (Mg = 1900 mg kg^−1^), and sulfur (S = 2900 mg kg^−1^). Meneses et al. [127] pointed out a sodium (Na) content of 137 mg kg^−1^, according to Ikram et al. [105]. The negative impacts of high Na levels on plant metabolism include cellular dehydration, nutrient deficiency, growth inhibition, stomatal rupture, and plant death [135].

Chemically, 100 g d.w. BSG contains about 15–25 g protein, 50–70 g fiber (hemicellulose, cellulose, and lignin), and 5–10 g fat [136].

Bachmann et al. [131] reported a cellulose content between 16.8 and 26.0% and 19.2 and 41.9% of hemicellulose. Lignin content was reported in a range of 15.4 [137]–28 g 100 g^−1^ [138]. Bachmann et al. [131] argue that a high lignin content makes BSG interesting for obtaining biochar. The high protein content is mainly characterized by the abundant presence of amino acids (2571.7 mg kg^−1^) [117,139]. The presence of lysine, histidine, and arginine is linked to anti-inflammatory potential and antimicrobial properties [140]. Belardi et al. [129] investigated the total polyphenols content, which reported 7.41 mg GAE (gallic acid equivalent) g^−1^ d.m. Niabaho et al. [126] reported that the type of grain, production process, sun exposure, soil type, and climatic conditions caused differences in the amounts of phenolic compounds. Table 6 shows the chemical composition of brewer’s spent grain, including nitrogen content, carbon content, and pH, as reported by several authors.

#### 6.3.3. Practical Applications

Brewer’s spent grain, with its high-water retention capacity, significant organic matter content, and richness in essential macronutrients (such as P, Ca, Mg, and S), presents as a promising organic component for growing substrates. However, its high moisture content and rapid degradability necessitate careful management to prevent spoilage, suggesting that preliminary treatments like drying or composting are required before incorporation into substrates. Furthermore, its lignocellulosic composition and high lignin content make it of interest for biochar production, which could further enhance its properties as a long-term amendment.

### 6.4. Crop’s Performance

According to Assandri et al. [141], the use of brewer’s spent grain (BSG) for agricultural purposes began to be studied in the early 90s, following the research of Mbagwu and Ekwealor [142] who used it as a fertilizer when combined with mineral fertilizers on maize. Subsequently, Yoo et al. [111] studied BSG-based biochar as an organic soil amendment for lettuce in response to soil acidification. The 5% application rate yielded the best lettuce fresh shoot weight increase. Cacace et al. [112] also examined its agronomic potential as an organic soil amendment in the escarole (*Cichorium endivia* var. Cuartana) cultivation. Results showed that BSG amendments increased leaf number and fresh/dry weight, soil organic carbon, total nitrogen, and water retention. Ebido et al. [113] evaluated BSG composted with poultry manure for amaranth (*Amaranthus viridis* L.) grown, which significantly increased total fresh biomass (143%) and dry matter (58%) compared to the control, correlated with increased exchangeable Ca and total N. Finally, Sdao et al. [57] evaluated BSG in container-grown ornamental marigold and nasturtium, at 10, 20, and 40% peat replacement. In marigold seedlings, 10% BSG improved chlorophyll content and photosynthesis, while 40% BSG led to growth depression. In marketable plants, 10% BSG produced root length and dry matter similar to peat control. Meanwhile, 20% and 40% doses reduced plant height and flower production due to increased electrical conductivity and phosphorus deficiency. The researchers concluded that BSG is a potential partial P substitute at low replacement rates. Table 7 compares peat, wood fiber, coffee silverskin, and brewer’s spent grain. It summarizes their origin, main advantages, and limitations as growing media components, along with relevant bibliographic references.

## 7. Conclusions

This systematic review highlights the significant potential of wood fiber, coffee silverskin, and brewer’s spent grain as viable components for horticultural growing media, offering sustainable alternatives to peat. While many of the findings presented in this review are interesting for practical applications, several research gaps and key technological challenges remain for their widespread and optimized adoption. For wood fiber, a well-established component, optimizing its inclusion rates for different species and overcoming potential water and nutrient availability limitations at higher concentrations are key research avenues. Long-term degradation patterns of wood fiber in growing media and strategies for improved nitrogen management and pH stabilization also warrant further investigation. Although coffee silverskin shows promising water retention and high nutrient content, the technological challenge of eliminating its intrinsic phytotoxicity at a large scale is key for broader application. Future research must fully elucidate specific inhibitory compounds and develop robust, industrial-scale pre-treatment methods. Similarly, for brewer’s grains, overcoming the obstacles of microbial content that leads to rapid spoilage and high salinity is critical to ensuring substrate performance and quality plants. Studies will also assess their integration into advanced cultivation systems such as hydroponics, aeroponics, and vertical farming, emphasizing enhancing sustainability and efficiency.

A central goal remains ensuring the environmental sustainability of wood fiber production relative to peat. Future research needs to go beyond the morphophysiological quality to truly confirm the use of these innovative organic matrices as viable peat alternatives. It also needs to assess their agronomic, environmental, and economic sustainability comprehensively. This means conducting Life Cycle Assessments (LCA) for the entire production process of plants grown in these new organic matrices, and thoroughly analyzing the production costs of the plant products. This will provide a complete and holistic picture of their sustainability compared to traditional peat-based cultivation.

## Figures and Tables

**Figure 1 plants-14-01945-f001:**
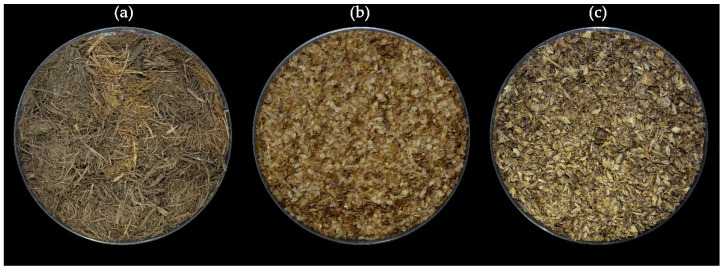
Wood fiber (**a**), coffee silverskin (**b**), and brewer’s spent grain (**c**).

**Figure 2 plants-14-01945-f002:**
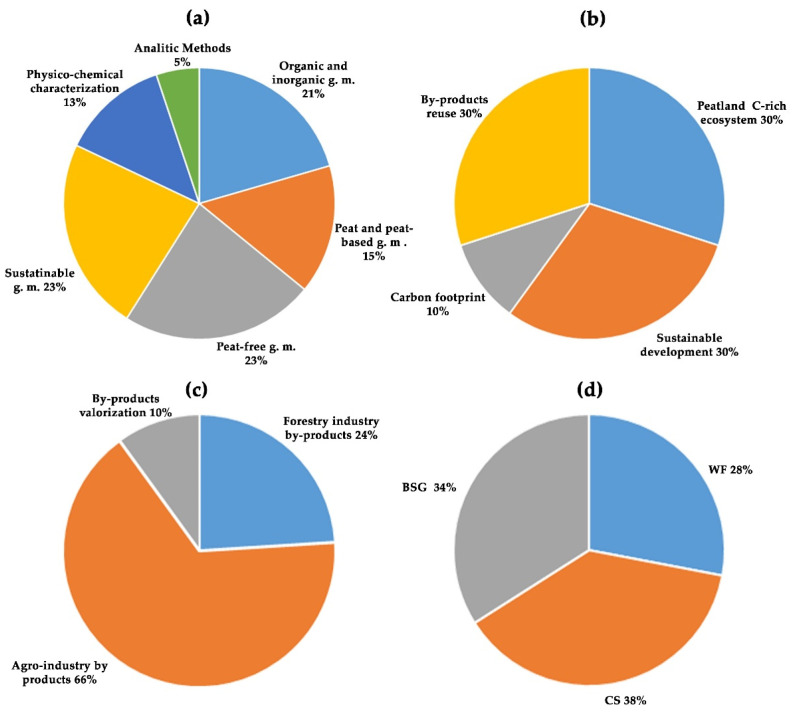
Keywords references (%) divided into categories of interest: (**a**) potted growing media; (**b**) agro-environmental sustainability; (**c**) forestry and agro-industry byproducts; (**d**) conventional and unconventional organic matrices as wood fiber (WF), coffee silverskin (CS), and brewer’s spent grain (BSG).

**Figure 3 plants-14-01945-f003:**
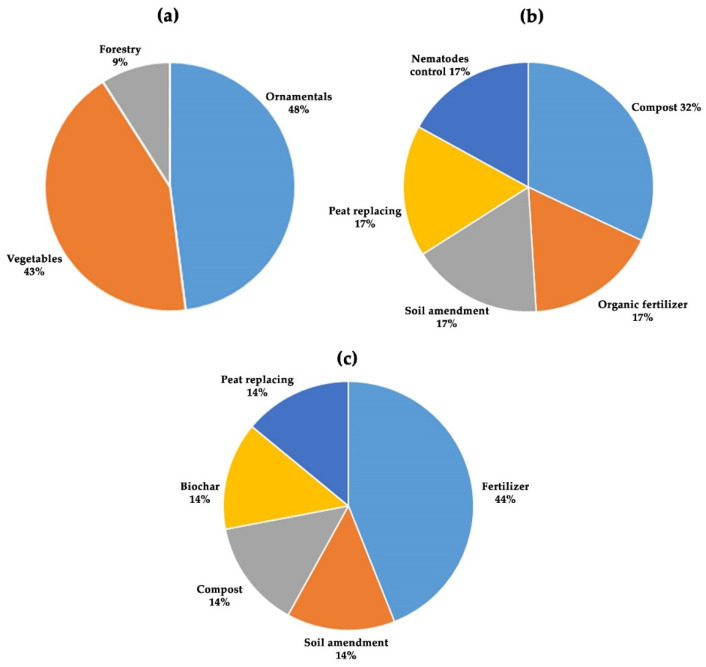
Overview of agricultural applications for the three organic materials. (**a**) Wood fiber (WF) is used predominantly in horticultural practices, with 91% applied to ornamental plants and vegetable production; (**b**) presents data on coffee silverskin (CS), where soil amendments and composting account for 49% of its agricultural uses, and peat replacement represents 17%; (**c**) shows that brewer’s spent grain (BSG) is primarily utilized as a fertilizer, making up 44% of its reported agricultural application.

**Table 1 plants-14-01945-t001:** Wood fiber physical and hydrological characterizations.

Parameters	Unit of Measure	Value	References
B.D.	g cm^−3^	0.07	[52]
P.D.	g cm^−3^	1.56	[52]
M.	%	10.3–41.0	[41,53]
T.P.S.	%	>85	[54]
E.A.W.	vol. %	13.8	[52]

B.D. = Bulk Density; P.D. = Particle Density; M = Moisture; T.P.S. = Total Pore Space; E.A.W. = Readily Available Water.

**Table 2 plants-14-01945-t002:** Wood fiber chemical characterization.

Parameters	Unit of Measure	Value	References
pH		4.1–6.0	[30,53]
E.C.	mS cm^−1^	4.0–28	[30,53,57]
C.E.C.	meq g^−1^	0.22	[53]
T.O.M.	%	95–98	[30,53]
O.C.	%	64–67	[30]
T.N.	%	0.20–0.45	[30,41,53,57]
C/N		168–456	[30,41,52]
P	mg kg^−1^	57	[41]
K	mg kg^−1^	472	[41]
Ca	mg kg^−1^	1574	[53]
Mg	mg kg^−1^	346	[53]
Fe	mg kg^−1^	46.5	[53]
Lignin	g kg^−1^ d.w.	360	[28]
Cellulose	g kg^−1^ d.w.	180	[52]
Hemicellulose	mg kg^−1^ d.w.	489	[52]

E.C. stands for Electrical Conductivity; C.E.C. stands for Cation Exchange Capacity; T.O.M. stands for Total Organic Matter; O.C. stands for Organic Carbon, T.N. stands for Total Nitrogen; and C/N stands for Carbon–Nitrogen ratio.

**Table 3 plants-14-01945-t003:** Coffee silverkin physical and hydrological characterization.

Parameters	Unit of Measure	Value	References
F.S.	mm	2–5	[94]
B.D.	g cm^−3^	0.18	[94]
P.D.	g cm^−3^	0.71	[94]
M.	%	7.1–7.3	[94,95,96]
W.H.C.	g H_2_O g^−1^ dry sample	5.11	[97]
W.R.	%	91.4	[97]

F.S. = Flake Size; B.D. = Bulk Density; P.D. = Particle Density; M. = Moisture content; W.H.C. = Water-Holding Capacity; W.R. = Water Retention.

**Table 4 plants-14-01945-t004:** Coffee silverskin chemical characterization.

Parameters	Unit of Measure	Value	References
pH		5.3–5.6	[78,98,99]
E.C.	dS m^−1^	1.8–4.9	[95,96,97]
Ash	g 100 g^−1^	5–7	[95,96,97,98]
T.O.M.	g 100 g^−1^	43.3–93.4	[83,85,98]
T.N.	g 100 g^−1^	0.14–5.1	[57,83,85]
C/N		11–19.1	[83,85,97]
P	g 100 g^−1^	0.76	[57]
K	g kg^−1^	7.6	[57]
Lignin	g 100 g^−1^	28.6–30.2	[96,100]
Cellulose	g 100 g^−1^	23.7	[96]
Hemicellulose	g 100 g^−1^	62.1	[96]
Protein	g 100 g^−1^	16.2–18.7	[96,100]
T.P.	mg kg^−1^	140	[101]

E.C. = Electrical conductivity; T.O.M. = Total Organic Matter; T.N. = Total Nitrogen; C/N = Carbon–Nitrogen ratio; T.P.= Total Polyphenol.

**Table 5 plants-14-01945-t005:** Brewer’s spent grain physical and hydrological characterization.

Parameters	Unit of Measure	Value	References
B.D.	g cm^−3^	0.129–0.159	[126]
M.	%	78–84	[112,127,128,129]
W.R.	%	207	[57]

B.D. = Bulk Density; M. = Moisture Content; W.R. = Water Retention.

**Table 6 plants-14-01945-t006:** Brewer’s spent grain chemical characterization.

Parameters	Unit of Measurement	Value	References
pH		4.5–6.0	[98,113,128]
E.C.	dS m^−1^	3480	[57]
Ash	%	1.9–8.5	[128,129,132,133]
T.O.M.	g 100 g^−1^	93.8	[57]
O.C.	%	47.3–53.0	[132,133,134]
T.N.	%	3.8–5.1	[132,133,134]
C/N		10.8–13.9	[132,133,134]
P	mg kg^−1^	6000	[127]
Ca	mg kg^−1^	3600	[127]
Mg	mg kg^−1^	1900	[127]
S	mg kg^−1^	2900	[127]
Na	mg kg^−1^	137	[105,127]
Lignin	g 100 g^−1^	15.4–28.0	[137,138]
Cellulose	%	16.8–26.0	[131]
Hemicellulose	%	19.2–41.9	[131]
Protein	g 100 g^−1^ d.w.	15–25	[136]
Amino acids	mg kg^−1^	2571.7	[117,139]
Fiber	g 100 g^−1^ d.w.	50–70	[136]
Fat	g 100 g^−1^ d.w.	5–10	[136]
T.P.	mg GAE g^−1^ d.m.	7.41	[129]

E.C. = Electrical conductivity; T.O.M. = Total Organic Matter; O.C. = Organic Carbon content; T.N. = Total Nitrogen; C/N= Carbon–Nitrogen ratio; T.P.= Total Polyphenol.

**Table 7 plants-14-01945-t007:** Comparative Analysis of *Sphagnum* Peat, Coffee Silverskin, Brewer’s Spent Grain, and Wood Fiber as Alternative Organic Materials for Growing Media.

Material	Origin	Advantages	Disadvantages	References
*Sphagnum* peat	Surface of the rewetted peat soils	Good T.P.S., air content, and W.H.C. The acidic pH value can easily be adjusted.	Contain herbal or grass seeds, poorly re-wettable; investment costs are still high.	[8]
Wood fiber	Byproducts from the wood industry (sawdust, chips, bark).	Good drainability, re-wetting. Low B.D. is used to optimize the physical properties of the blend. It reduces B.D., increases airspace, and improves rewettability.	May cause nitrogen immobilization (due to degradation by microorganisms). High C/N. Degradation by microorganisms leads to deterioration of physical properties. Relatively low N content. It can impair W.R. and nitrogen availability if used in excessive proportions. High proportions (≥75%) can lead to chlorosis and reduced growth unless pH and nutrient supply are managed.	[4,5,8,12,25,27,28,29,30,31,32,33,34,35,36,37,38,39,40,41,42,43,44,45,46,47,48,49,50,51,52,53,54,55,56,57,58,59,60,61,62]
Coffee silverskin	Byproduct of coffee roasting.	Rich in nutrients (N, P, K, etc.). High W.R. capacity, high O.M., acidic pH. Antioxidant properties. Compostable to eliminate phytotoxicity.	Phytotoxicity (polyphenols, chlorogenic acid) inhibits germination. Low C/N. Compositional variability. Low nitrogen mineralization.	[57,68,69,70,71,72,73,74,75,76,77,78,79,80,81,82,83,84,85,86,87,88,89,90,91,92,93,94,95,96,97,98,99,100,101,102]
Brewer’s spent grain	Byproduct of brewing (mashing).	Rich in fiber, protein, and bioactives (amino acids). Biochar, soil conditioner, compost, and peat substitute (low doses) are used in agriculture. Good W.R. Sub-acid pH.	High M. and sugar content (rapid deterioration: 7–10 days); needs refrigeration/drying. High Na levels (negative impact on plants). At high doses, it reduces growth and flowering due to high EC and P deficiency.	[57,98,104,105,106,107,108,109,110,111,112,113,114,115,116,117,118,119,120,121,122,123,124,125,126,127,128,129,130,131,132,133,134,135,136,137,138,139,140,141,142,143]

T.P.S. = Total Pore Space; W.H.C. = Water-Holding Capacity; B.D. = Bulk Density, W.R. = Water Retention; O.M. = Organic Matter content; M. = Moisture content; E.C. = Electrical conductivity; P = Phosphorous.

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
