# Peer review of "Beyond Peat: Wood Fiber and Two Novel Organic Byproducts as Growing Media—A Systematic Review"

_plants, 2025, doi:10.3390/plants14131945_

Round 1
Reviewer 1 Report
Comments and Suggestions for Authors
OBSERVATIONS (Beyond Peat: Wood Fiber and Two Novel Organic By-products as Growing Media – A Systematic Review).
General comments
I consider in general this is a well written manuscript. This is a review I consider that can be publishable. Manuscript seems to me interesting based on the benefit from reducing significantly the use of peat for seedling production using organic by-products as part of the substrates components.
This review in its current format, however, there is a missed opportunity to provide a more comprehensive discussion of growing media. Some things that could be included as part of the crop´s performance are little more background information about the role of growing media for nutrition and root development, the cost of growing media, and other interesting factors. Obviously, those things will vary geographically, but they are worth including.
Keywords: I suggest change wood fiber for sawdust or other term, since wood fiber is already included in the title.
Specific comments in the text:
Page 3 line 99, I suggest “by-products” in order to be consistent in the text from the complete document.
Page 3 line 110, I suggest split this paragraph starting at: Several standard growing media… and include some cites at the end of this new paragraph to support information of the three mentioned materials.
Page 3 line 131, Eliminate point before the word Methodology.
Page 6 line 206, I suggest include one or two more cites to support seedling production of forest shrubs and trees in order to have a better balance compared to vegetables and ornamentals. For example:
González-Orozco, M. M., J. Á. Prieto-Ruíz, A. Aldrete, J. C. Hernández-Díaz, J. A. Chávez-Simental, and R. Rodríguez-Laguna. 2018. Nursery production of Pinus engelmannii Carr. with substrates based on fresh sawdust. Forests 9 (11), 678. DOI: https://doi.org/10.3390/f9110678
Page 7 and 8 lines 251-296, I suggest split this long paragraph in order to make it more understandable.
Page 9 line 338, “density of 0.18 g cm-³ and a particle density of 0.71 g/cm³”, look for consistency in the units. Furthermore, review all this section to correct super indices in lines 344, 346, 352, 353, 354. Review also signs (-1).
Page 11 lines 412-413, review scientific names italics.
Page 11 line 457, review units (0.129-0.159 g cm-3).
Comments on the Quality of English LanguageThe English is fine in general, but could be improved if a native English speaker review the document.
Author Response
1 Provide a more comprehensive discussion of growing media
We appreciate the reviewers' insightful feedback. In response to the comments and suggestions, we have rewritten a large part of our manuscript to enhance clarity and address the concerns raised. We believe these amendments significantly improve the overall quality of the work.
2
|
crop´s performance are little more background information: the role of growing media for nutrition and root development |
Thank you for the suggestion. We have added a brief background on how growing media influence nutrient availability and root development, which are key factors in crop performance |
Lines 83-86 |
3
|
the cost of growing media |
Amended: We have added a short section discussing the cost of different growing media and their implications for production economics |
lines87-91 |
4
|
Keywords: I suggest change wood fiber for sawdust |
Thank you! Done |
Lines 29 |
5
|
Page 3 line 99, I suggest “by-products” in order to be consistent in the text from the complete document.
|
Thank you! done |
Lines 112 |
6
|
On page 3, line 110, I suggest splitting this paragraph starting at Several standard growing media… and including some cites at the end of this new paragraph to support the information about the three mentioned materials.
|
We have split the paragraph at the recommended point and added relevant citations to support the information on the three mentioned materials |
Lines 125-152 |
7
|
Page 3 line 131, Eliminate point before the word Methodology.
|
Thank you! done |
Line 153-154 |
8
|
Page 6 line 206, I suggest include one or two more cites to support seedling production of forest shrubs and trees in order to have a better balance compared to vegetables and ornamentals |
Done : González-Orozco, M. M., J. Á. Prieto-Ruíz, A. Aldrete, J. C. Hernández-Díaz, J. A. Chávez-Simental, and R. Rodríguez-Laguna. 2018. Nursery production of Pinus engelmannii Carr. with substrates based on fresh sawdust. Forests 9 (11), 678. DOI: https://doi.org/10.3390/f9110678
|
Reference n. 16 Lines 88 |
9
|
Page 7 and 8 lines 251-296, I suggest split this long paragraph in order to make it more understandable.
|
Thank you! Done |
Lines 239-243 245- 274 281-301 |
10
|
Page 9 line 338, “density of 0.18 g cm-³ and a particle density of 0.71 g/cm³”, look for consistency in the units. Furthermore, review all this section to correct super indices in lines 344, 346, 352, 353, 354. Review also signs (-1).
|
Thank you! Done. |
Lines 449 |
11
|
Page 11 lines 412-413, review scientific names italics.
|
Thank you! Done. |
Line 399,400,402 |
12
|
Page 11 line 457, review units (0.129-0.159 g cm-3). |
Thank you! Done. |
Line 612 |
13
|
The English is fine in general, but could be improved if a native English speaker reviews the document.
|
Thank you for the observation. We have carefully revised the manuscript to improve clarity and flow. |

Reviewer 2 Report
Comments and Suggestions for Authors
Dear Authors,
The submitted work is a review article that provides an overview of the physical, chemical, and biological characteristics of growing media used in soilless cultivation systems. The topic is both timely and relevant, given the increasing global interest in sustainable horticultural practices and resource-efficient food production. The manuscript covers a broad spectrum of substrates, including organic and inorganic materials, and examines their interactions with plant growth, microbial dynamics, and environmental effects. The inclusion of recent innovations and sustainability considerations adds further value to the article. Nevertheless, the structure of the manuscript would benefit from clearer thematic subdivisions and the inclusion of a more coherent conceptual framework linking substrate properties to practical applications. Certain sections are overly descriptive and would be strengthened by a more critical synthesis of the literature. The addition of comparative tables or illustrative figures is recommended to enhance clarity and readability. The reference list is generally adequate; however, several recent studies (post-2022) concerning bio-based and circular-economy-derived substrates appear to be missing and should be incorporated. Moreover, the conclusions would be more impactful if they emphasized key research gaps and outlined technological challenges for future investigation. In summary, the manuscript addresses an important and current topic. With revisions aimed at improving structure, critical depth, and visual presentation, the article has the potential to make a meaningful scholarly contribution to the field.Author Response
Reviewr no. 2
|
Question |
Answer |
lines |
|
Reviewer no 2 |
|
|
|
1 clearer thematic subdivisions and the inclusion of a more coherent conceptual framework linking substrate properties to practical applications.
|
Done: a paragraph titled “Practical applications” was added to the manuscript for each organic by-product |
Lines 310-317 508-515 659-667 |
|
2 Certain sections are overly descriptive and would be strengthened by a more critical synthesis of the literature.
|
Thank you for the valuable feedback. In response, we have substantially rewritten the manuscript to reduce overly descriptive sections and provide a more critical synthesis of the literature throughout. |
|
|
3 The addition of comparative tables or illustrative figures is recommended to enhance clarity and readability. |
Done Figure 1. – Wood fiber (a), coffee silverskin (b) and brewer’s spent grain; Figure 2. Keywords references (%) divided into categories of interest: (a) potted growing media; (b) agro-environmental sustainability; (c) forestry and agro-industry by-products; (d) conventional and unconventional organic matrices as WF, CS, and BSG. Figure 3. Overview of agricultural applications for the three organic materials. (a), Wood fibre (WF) is used predominantly in horticultural practices, with 91% applied to ornamental plants and vegetable production. (b) presents data on coffee silverskin (CS), where soil amendments and composting account for 49% of its agricultural uses, and peat replacement represents 17%. (c) shows that brewer’s spent grain (BSG) is primarily utilized as a fertilizer, making up 44% of its reported agricultural application. Table 1. Wood fiber physical and hydrological characterization. Table 2. Wood fiber chemical characterization. Table 3. Coffee silverkin physical and hydrological characterization. Table 4. Coffee silverskin chemical characterization. Table 5. Brewer’s spent grain physical and hydrological characterization. Table 6. Brewer’s spent grain chemical characterization. Table 7. Comparative Analysis of Coffee Silverskin, Brewer's Spent Grain, and Wood Fiber as Alternative Organic Materials for Growing Media.
|
Lines 130 183 189 277 303 456 492 622 655 710 |
|
4. Several recent studies (post-2022) concerning bio-based and circular-economy-derived substrates appear to be missing and should be incorporated |
Done: we added references no. 73,79,110 |
Lines 409,422, 591 |
|
5. The conclusions would be more impactful if they emphasized key research gaps and outlined technological challenges for future investigation |
We have revised the conclusion to highlight key research gaps and outline major technological challenges that warrant further investigation |
Lines 719 |

Reviewer 3 Report
Comments and Suggestions for Authors
This a thorough review of peat alternatives including wood fiber, coffee silverskin and brewer’s spent grain. This work provides useful information for soilless culture and the substrate industry. However, I have a couple of comments which may help improve the quality of this manuscript:
- My primary concern is the readability of this manuscript. It is quite lengthy and full of very long paragraphs. For example, part 3.3 comprised a single paragraph of more than 40 lines. The author may want to break it into shorter paragraphs.
- This manuscript is mostly narrative and lacks summarization. The authors simply cited many literatures and described what they do pe the results. I strongly encourage the authors to further summarize the literature using tables or figures, which may improve the readability. By far this review only contains one table.
- The flow structure should be more logical. For example, “Physical, chemical, and hydrological characterization” and “crop performance” need to be modified. In crop performance, for example, line 257-259, the authors talked about porosity, which I believe is a physical property.
- Alignment of Figure 1 and 2 needs to be modified. They occupy a large space, and size of each figure was not consistent.
- Line 171: I am confused about this sentence. “2017 3 Mm3y-1 to 30 Mm3y-1 by 2050”, please double check if the unit is correct.
Author Response
Reviewer no.3
|
Question |
Answer |
lines |
|
Reviewer no 3 |
|
|
|
1.full of very long paragraphs. For example, part 3.3 comprised a single paragraph of more than 40 lines. The author may want to break it into shorter paragraphs |
Done: we had created sub-paragraphs as: - Physical, hydrological and chemical characterization - Physical and hydrological characterization - Chemical characterization |
Lines 239 245 282
|
|
2. The manuscript lacks summarization |
Thank you for pointing this out. We have added summarizing statements throughout the manuscript, particularly at the end of key sections, to improve clarity and cohesion |
|
|
3. summarize the literature using tables or figures, |
We added 7 Tables that summarize the references |
Lines 277 303 456 492 622 655 710
|
|
4. The flow structure should be more logical |
We have reorganized the manuscript to improve the logical flow and coherence of the content. |
|
|
5. In crop performance, for example, line 257-259, the authors talked about porosity, which I believe is a physical property.
|
Thank you for the clarification. We have revised the text at lines 257–259 (old manuscript) to clearly specify porosity as a physical property of the growing media. |
Lines 329 |
|
6. Alignment of Figure 1 and 2 needs to be modified. They occupy a large space, and size of each figure was not consistent.
|
We have adjusted the alignment and resized Figures to ensure consistency and better use of space in the manuscript. |
Lines 180 188 |
|
7. Line 171: I am confused about this sentence. “2017 3 Mm3y-1 to 30 Mm3y-1 by 2050”, please double check if the unit is correct.
|
Thank you. Done |
Lines 198 |

Round 2
Reviewer 3 Report
Comments and Suggestions for Authors
The authors have made substantial revision to this manuscript. I believe that my concerns have been addressed and I have no more comments.